# Calcium-Dependent Pulmonary Inflammation and Pharmacological Interventions and Mediators

**DOI:** 10.3390/biology10101053

**Published:** 2021-10-16

**Authors:** Jeffrey G. Shipman, Rob U. Onyenwoke, Vijay Sivaraman

**Affiliations:** 1Department of Biological and Biomedical Sciences, College of Health and Sciences, North Carolina Central University, Durham, NC 27707, USA; jshipma7@eagles.nccu.edu; 2Department of Pharmaceutical Sciences, North Carolina Central University, Durham, NC 27707, USA; ronyenwo@nccu.edu; 3Biomanufacturing Research Institute and Technology Enterprise (BRITE), North Carolina Central University, Durham, NC 27707, USA

**Keywords:** calcium (Ca^2+^), inflammation, pulmonary biology, macrophages, smoking, vaping, pathology

## Abstract

**Simple Summary:**

Pulmonary diseases such as asthma, chronic obstructive pulmonary disease (COPD) and acute respiratory disease syndrome (ARDS) are common throughout the world. Tobacco products can potentially lead to lung damage, which in turn contribute to worsening disease outcomes. The same can be said of E-cigarette (E-cig) use, which has recently gained attention due to its possible adverse side effects. Despite this information, little is known about the potential for causality between smoking and E-cig use, and lung damage and worsening disease outcomes. In this review, we focus on a potential inflammatory mechanism, that is, calcium (Ca^2+^) signaling and its importance in many normal biological processes as well as roles it may play in the adverse effects of tobacco and E-cig use. In addition, we discuss possible therapeutic small molecules that may improve pulmonary disease outcomes in the future.

**Abstract:**

Pulmonary diseases present a significant burden worldwide and lead to severe morbidity and mortality. Lung inflammation caused by interactions with either viruses, bacteria or fungi is a prominent characteristic of many pulmonary diseases. Tobacco smoke and E-cig use (“vaping”) are considered major risk factors in the development of pulmonary disease as well as worsening disease prognosis. However, at present, relatively little is known about the mechanistic actions by which smoking and vaping may worsen the disease. One theory suggests that long-term vaping leads to Ca^2+^ signaling dysregulation. Ca^2+^ is an important secondary messenger in signal transduction. Cellular Ca^2+^ concentrations are mediated by a complex series of pumps, channels, transporters and exchangers that are responsible for triggering various intracellular processes such as cell death, proliferation and secretion. In this review, we provide a detailed understating of the complex series of components that mediate Ca^2+^ signaling and how their dysfunction may result in pulmonary disease. Furthermore, we summarize the recent literature investigating the negative effects of smoking and vaping on pulmonary disease, cell toxicity and Ca^2+^ signaling. Finally, we summarize Ca^2+^-mediated pharmacological interventions that could potentially lead to novel treatments for pulmonary diseases.

## 1. Introduction

Respiratory diseases such as pneumonia, acute respiratory disease syndrome (ARDS) [1] and chronic obstructive pulmonary disease (COPD) [2] carry a significant worldwide burden of morbidity and mortality. Pulmonary diseases are characterized by inflammation within one or both lungs and are typically caused by infection with viruses, bacteria or fungal pathogens [3,4]. The stages of pulmonary inflammation are typically initiated through the interaction of the microbe/microbial product with the airway epithelium. This interaction leads to a pro-inflammatory response mediated by cytokines and chemokines as well as macrophages and activated neutrophils at the site of insult [5,6]. In most cases, a healthy immune response is able to clear the infection, and the inflammation resolves. However, certain pathologic conditions result in robust neutrophil activation, leading to pneumonia and other forms of acute lung injury (ALI) such as ARDS. 

Tobacco smoking is one of the major risk factors for the development of pulmonary disease and dysfunction [2]. In the US alone, tobacco use has accounted for ~20 million premature deaths since the publication of the Surgeon General’s report in 1964. Tobacco exposure triggers a number of inflammatory responses in the airways, which can manifest as a number of very diverse ailments and include more commonly thought of pulmonary ailments: airway inflammation, ALI, ARDS, COPD and lung cancer as well as inflammatory bowel diseases, which often present along with pulmonary diseases such as COPD [2,7,8] At the same time, tobacco smoke and the use of electronic cigarettes (or “vaping”) itself renders the smokers’ and vapers’ lungs more susceptible to microbial infection/burden and, therefore, to pulmonary disease [9,10]. Because smokers’ airways are characterized by dehydrated mucus, many well-documented studies have investigated cystic fibrosis transmembrane conductance regulator-mediated regulation [11] and the critical role that calcium (Ca^2+^) signaling and, in particular, abnormal Ca^2+^ influx plays in cystic fibrosis [12,13,14,15] Both acute and chronic exposure to cigarette smoke lead to a chronic elevation in intracellular Ca^2+^ levels, and smoke from similar tobacco products is expected to exert similar effects, resulting in pulmonary disease initiation and progression [12,13,16]. Although Ca^2+^ dysregulation was shown to play a pivotal role in inflammation and to result in acute pulmonary disease, little is known about the mechanisms of Ca^2+^-induced inflammatory lung diseases within the context of epithelial and endothelial cell function. Furthermore, abnormal Ca^2+^ homeostasis has itself been linked to the same pathologies as tobacco smoke, i.e., inflammation, ALI, ARDS, COPD and lung cancer [17]. 

As can be inferred from above, tobacco smoke exerts multiple effects on the airway epithelia and elevates intracellular Ca^2+^ ([Ca^2+^]_i_), leading to dysfunctional ion secretion. When agonists bind to ion channels, the channels open to allow the flow of cation ions (Na^+^, Ca^2+^, K^+^) through the cell membrane, inducing a wide variety of biological responses. Given the large number of physiological processes governed by changes in [Ca^2+^]_i_, Ca^2+^ signaling clearly plays a pivotal role in pulmonary disease [12]. In fact, a sustained elevation of the basal Ca^2+^ concentration likely leads to cell stress, activating autophagy and cell death [18,19]. Herein, we will provide a detailed understanding of (1) the role of Ca^2+^ and Ca^2+^ signaling under basal physiological conditions, and (2) Ca^2+^ dysregulation as a mechanism of pulmonary disease development and progression with specific emphasis on tobacco products, which we focus upon due to their status as the major risk factor for pulmonary disease development.

## 2. Ca^2+^ and Signal Transduction

Ca^2+^ is an important secondary messenger within cells, triggering many cellular processes. Various cellular compartments utilize and manipulate Ca^2+^ concentrations. In terms of complexity, there exists a series of pumps, channels, transporters and exchangers that mediate intracellular and extracellular Ca^2+^ concentrations [20]. This complex system allows for the manipulation of Ca^2+^ to trigger numerous intracellular processes including cell metabolism, proliferation, death, gene transcription, secretion, etc. [20,21]. For example, Ca^2+^ mobilization generates signals based on its release from endoplasmic reticulum (ER) stores and the influx of extracellular Ca^2+^ [22]. Ca^2+^ signaling utilizes multiple channels to mediate the cellular uptake of extracellular Ca^2+^ into cells. Among the most important of these channels are the Transient receptor protein channels (TRP), Store Operated Calcium Entry (SOCE) channels and Voltage-Gated Calcium (VGC) channels [20]. Thus, the activation of these channels is a major factor in regulating the uptake of Ca^2+^ into the cytosol of non-neuronal cells.

### 2.1. Transient Receptor Potential Channels

Transient receptor potential channels (TRP) are encoded in humans via six different orthologous genes (TRPC1-7 with the exclusion of TRPC6) [23]. TRP channels can be activated by depletion of internal stores of Ca^2+^ and exhibit significant diversity in terms of specificity and mechanisms of activation compared with other ion channels. These channels allow cells to respond to external stimuli as well as to sense changes in the local environment. There are many subfamilies of TRP channels that are activated and regulated in different ways. However, all TRPs share common activation cues through pathways that couple phospholipase C (PLC). Humans express six of the seven TRPC proteins due to *TRPC2* being a pseudogene [24]. All human TRPs use PLC for activation, though there is variability in specificity for Ca^2+^ and other cations. For example, many TRPs are activated upon the depletion of internal Ca^2+^ stores through IP_3_ or thapsigargin [23]. Interactions between ER Ca^2+^ release channels and TRPs have previously been described. However, TRP channels may still be active even after ER-resident Ca^2+^ channels, i.e., IP_3_R homologs, are knocked out, which suggests another mechanism of activation [8]. Thus, the current literature hypothesizes a complex relationship between Ca^2+^ store depletion and TRP activity and that a conformational coupling mechanism exists between TRPCs and ER-resident Ca^2+^ release channels. This mechanism would require a close but separate interaction between the ER and plasma membranes. Pharmacological reorganization of the actin cytoskeleton was also shown to decrease channel activity either by internalization or through the inactivation of exogenously expressed TRPC3 channels [23,25,26].

### 2.2. Store-Operated Calcium Entry

Store-operated Ca^2+^ entry (SOCE) is activated by the depletion of Ca^2+^ from ER-resident stores. In addition to refilling ER Ca^2+^ stores, SOCE channels also increase cytosolic Ca^2+^ levels to serve wider signaling functions [27,28]. The most prominent configuration for this channel is known as the Ca^2+^ release-activated Ca^2+^ (CRAC) current, which consists of the ER Ca^2+^ sensor STIM (Stromal Interaction Molecule) protein, and ORAI proteins, which are plasma membrane (PM) channels that are highly selective for increasing intracellular [Ca^2+^] [29]. STIM proteins sense Ca^2+^ in the ER lumen and communicate Ca^2+^ store depletion to other proteins such as ORAI [6]. STIM causes the channel pore within ORAI to open, allowing Ca^2+^ to flow through the plasma membrane and into the cytoplasm. STIM1 and STIM2 proteins function by way of a low-affinity EF-hand Ca^2+^ binding site. STIM1 forms discrete PM-associated junctions and physically associates with ORAI1 and other ORAI channels, which are selective channels located within the plasma membrane [30,31]. STIM1 and STIM2 homologs are type-I membrane proteins with a luminal helix-turn-helix motif EF-hand Ca^2+^-binding site as well as a sterile alpha motif (SAM) [31]. STIM1 localizes in the tubular ER but was detected in the plasma membrane where it is N-glycosylated. Evidence suggests that this N-glycosylation plays a role in plasma membrane delivery. Upon depletion of Ca^2+^ stores, STIM1 begins clustering and localizes to plasma membrane-adjacent ER regions [32]. This decrease in Ca^2+^ binding to the EF-hand specifically results in the clustering of STIM1 proteins followed by these multimers translocating to PM-adjacent ER areas where they can activate Ca^2+^ influx. ORAI proteins play a vital role in the CRAC current configuration. Three ORAI proteins also have varying functions in this pathway, which is despite their sequence similarity [31]. ORAI1 is thought to be most important in terms of Ca^2+^ influx into cells. Furthermore, decreasing its expression level has the most significant impact on SOCE [33,34]. In addition, ORAI1 and ORAI2 produce highly selective Ca^2+^ CRAC currents while ORAI3 produces small and developing CRAC currents [31,35]. The active pore of the channel consists of four ORAI molecules, which are induced by STIM1 interaction [31]. ORAI1 alone inhibits SOCE. Thus, the co-expression of STIM1 and ORAI1 is necessary to induce the influx of Ca^2+^ and the creation of CRAC channels [33,36].

### 2.3. Voltage-Gated Ca^2+^ Channels

Voltage-gated Ca^2+^ channels (VGCC) are activated upon the depolarization of the membrane and mediate Ca^2+^ entry into the cell. This event generates electrical signals that mediate many cellular events, e.g., enzyme activation and gene expression [37]. VGCCs are key signal transducers that convert the action potential signal in the membrane to a transient intracellular Ca^2+^ signal. These Ca^2+^ currents vary by cell type as defined by pharmacological criteria [15]. L-type Ca^2+^ currents exhibit slow voltage-dependent inactivation and, as a result, are long-lasting when (1) Ba^2+^ is the carrier and (2) there is no Ca^2+^-dependent inactivation. T-type Ca^2+^ channels are activated at more negative membrane potentials than L-type and are inactivated rapidly and deactivated slowly [38]. N-type Ca^2+^ channels have different intermediate voltage dependence and rate of inactivation when compared to L and N-type channels, which are more negative and faster than the L-type and more positive and slower than the T-type. Finally, there are L- and T-type Ca^2+^ currents, which are found in a variety of cells.

## 3. Additional Ca^2+^ Signaling Components

Additional proteins serve as integral channel components for Ca^2+^ release and signaling:

Ca^2+^ release through Inositol 1,4,5 triphosphate receptors (IP_3_R), which are located on the ER, can also drive increases in intracellular [Ca^2+^] [20,39]. IP_3_Rs are activated when cell surfaces receptors are activated [40]. Following activation, Ca^2+^ is released from the ER, Golgi apparatus and nuclear envelope, and IP_3_Rs localized to the plasma membrane are activated [41,42]. IP_3_ production is triggered by external stimuli that bind to cell surface receptors. These binding events induce Ca^2+^ signaling within cells over a short period of time [21,43]. The pathway for IP_3_ production and IP_3_R activation is referred to as the phosphoinositide signaling pathway. Phospholipase C (PLC), which interacts with many other Ca^2+^ signaling pathways, produces IP_3_ in cells and is expressed in different isoforms; however, only a few isoforms activate in response to external stimuli. Other isoforms may be activated by Ca^2+^, G protein Ras or are introduced during fertilization [21]. The hydrolysis of PIP2 by PLC not only produces IP_3_ but also diacylglycerol (DAG), which is maintained within the plasma membrane and plays a role in other Ca^2+^ channels as well as in the activation of protein kinase C. Conversely, the DAG may be metabolized to produce other cellular messengers.

Ryanodine receptors (RyRs) share 40% homology with IP_3_Rs and are Ca^2+^ release channels located on the sarcoplasmic reticulum and endoplasmic reticula [44]. In tissues, the influx of Ca^2+^ through plasma membrane Ca^2+^ channels activates the RyRs, resulting in rapid, massive releases of intracellular Ca^2+^ stores.

Sarcoendoplasmic reticulum Ca^2+^-ATPase (SERCA) is responsible for pumping Ca^2+^ into the ER lumen. These receptors mediate the storage of Ca^2+^ in the ER. Voltage-dependent anion channels (VDAC) and the mitochondrial Ca^2+^ uniporter family mediate mitochondrial Ca^2+^ entry [20]. Figure 1 [20] illustrates the diversity of pathways mediated by intracellular Ca^2+^ signaling.

Ca^2+^ signaling is not only important in intracellular processes but also plays a large role in immune cell response and inflammation. Ca^2+^ plays a pivotal role in cells as a secondary messenger. Stimulation of Ca^2+^ is responsible for the activation and regulation of many immune cells. Macrophages are key immune cells that function through phagocytosis. Their phagocytic function is dependent on internal Ca^2+^ regulation [45]. Increases in cytosolic Ca^2+^ are important for phagocytosis as well as phagosome maturation. In addition, Ca^2+^ binding to phagocytic receptors on macrophages triggers IP_3_ production. This release of Ca^2+^ from the ER triggers SOCE. The influx of extracellular Ca^2+^ is also required for macrophage polarization into the M1 state, which is pro-inflammatory and important for fighting pulmonary infections. Lowering the cytosolic Ca^2+^ concentration can modulate the switch of M1 macrophages to M2, which are termed anti-inflammatory.

In neutrophils, Ca^2+^ signaling is regulated mainly by SOCE channels. The influx of Ca^2+^ into the cytosol is responsible for neutrophil activation. In neutrophils, SOCE is initiated by ligand-receptor interactions, which trigger a complex cascade of molecules and results in ORAI1 stimulation. Stimulation of ORAI1 via STIM1 leads to Ca^2+^ influx, which increases the cytosolic Ca^2+^ concentration [46]. This results in the pro-inflammatory activation of neutrophils.

In lymphocytes, Ca^2+^ is present at low concentrations prior to activation [47]. Cross-linking of antigen receptors leads to the activation of phospholipase C, which breaks down phosphatidylinositol-4-5-biphosphate to create inositol-1,4,5-trisphosphate (IP_3_) and diacylglycerol. IP_3_ binds to receptors, leading to the release of Ca^2+^ into the cytosol. This depletion of the internal Ca^2+^ store activates SOCE channels, which are lymphocytes’ main mode of Ca^2+^ influx. Ca^2+^ release-activated channels (CRAC) are the most common channel configuration for SOCE found in lymphocytes.

Ca^2+^ dysregulation is an attributable trait that was described for various diseases including pulmonary diseases such as asthma and ALI [48]. However, little is known about the mechanistic action of Ca^2+^ dysregulation in these diseases.

## 4. Smoking and Ca^2+^ Derangement and Disease

COPD is an inflammatory lung disease, a major global health threat and is characterized by obstructed airflow within the lungs [49]. This obstruction of airways is progressive and often a result of toxic chemical exposure, leading to irregular inflammatory responses within the lungs [50]. This airway obstruction is not fully reversible and is caused by the progressive narrowing, destruction and removal of terminal bronchioles, which eventually results in emphysema [51]. ARDS is the result of a complex cascading process that develops as a result of ALI [52] and is characterized by acute respiratory failure because of increases in inflammation and pulmonary edema [5]. Many patients with ALI also exhibit cases of diffuse alveolar damage, which is an injury to the gas exchange surfaces within the lung and eventually results in damage to the blood–air barrier [53]. Asthma is an inflammatory lung disease characterized by pulmonary inflammation and airway obstruction, with the risk of developing asthma being higher among those who smoke and have smoked than those who have never smoked [54]. These various pulmonary diseases share the commonality that smoking is a major contributor to their resultant dysfunctional contraction profiles. Smoking and the use of other tobacco products present significant health risks and lead to these pulmonary diseases. Smoking exposes humans to numerous toxic chemicals, leading to worsened mobility and significant mortality due to the development of pulmonary diseases, as previously described. These activities result in inflammatory reactions within the airway, as well as alterations to innate immunity in the oral, nasal and airway passageways, which is in addition to changes in adaptive immunity [55].

Recently Ca^2+^ signaling dysfunction has gained attention as a potential cause for increased morbidity and disease mortality, with cytosolic Ca^2+^ concentrations increasing following smoking or the use of other tobacco products [12]. The study cited here demonstrated that Ca^2+^ influx, as well as endoplasmic reticulum and mitochondrial stores, do not play a role in the Ca^2+^ release caused by tobacco use. The study suggests that the Ca^2+^ originated from lysosomes and was maintained in the cytosol during chronic tobacco use. Total particulate matter (TPM) from tobacco products induced an increase in Ca^2+^ unrelated to extracellular Ca^2+^ and did not require the activation of the IP_3_ and/or TRP channel pathways. Another study [22] demonstrated that Ca^2+^ influx in bronchial epithelial cells was decreased in smokers’ lungs, which occurred simultaneously with a decrease in Ca^2+^ release from the ER because of the differential expression of ORAI3. This differential expression inhibited SOCE channels. As alluded to earlier, these data suggest that ORAI3 expression might play a large role in Ca^2+^ signaling impairment in smokers as well as in patients with COPD.

## 5. Emerging Tobacco Products and Ca^2+^ Derangement and Disease

Electronic cigarette (E-cig) use or “vaping” has become more prevalent in countries worldwide as it was initially portrayed as a safer alternative to cigarette smoking. Despite this information, little is known if long-term vaping will have similar effects to cigarette smoking. Recent studies have indicated possible adverse side effects associated with vaping [56,57]. For example, vaping is associated with increases in ALI and other pulmonary toxicities [58]. Vaping may expose humans to oxidant or reactive oxygen species (OX/ROS), which is suggested by the oxidant reactivity of E-cig aerosols [59]. OX/ROS are present in cigarette smoke and generate tars that lead to lung inflammation, which is the cause of many pulmonary diseases such as COPD, ALI and lung cancer [59]. The flavor chemicals contained within vaped e-liquid products may also play a determining role in any vaping-associated toxicity. Recent studies have utilized in vitro and in vivo models to better understand and study any potential interactions that these vaped aerosols may have within the lung and any resultant pulmonary toxicity. In vitro cell viability studies utilizing various lung cells have shown marked decreases in cell viability. Recent examples include studies using A549 (Human lung epithelial cells) and Calu3 (non-small-cell lung cancer), which are ideal models for studying the airway epithelium. These studies noted significant cytotoxic effects and significant decreases in cell viability when exposing these cells to different e-liquids possessing various flavors [60,61]. In particular, “Mint” flavored JUUL e-liquid decreased cell viability, led to elevated levels of the pro-inflammatory cytokine IL-6 and increased levels of intracellular Ca^2+^ and Annexin V, with Annexin V being a well-described marker for apoptosis [61]. Other examples include a combined effect with cinnamon flavorings and nicotine, with the combination leading to a significantly greater decrease in cell viability than cinnamon alone in A549 cells, suggesting that nicotine and flavor combinations may exhibit synergistic activities [60]. Cinnamon-flavored products also produce slight increases of the pro-inflammatory cytokine IL-8 and anti-inflammatory cytokine MCP-1. Bengalli and colleagues [60] utilized a co-culture of the human lung epithelial cell line NCI-H441 and human pulmonary microvascular endothelial cell line HPMEC-ST1.6R to mimic the alveolar–blood barrier. During smoking or vaping, these lung epithelial cells were exposed to aerosols, which altered the integrity of the barrier and resulted in ALI and diffuse alveolar damage. The results from this study revealed that vaping flavored e-liquid products, specifically cinnamon- and menthol-containing, significantly decreased the integrity of the barrier. Recent research studies have also begun to interrogate the Ca^2+^ dysregulation caused by e-liquids/vaping based upon observations of similar health impacts to those resulting from the use of tobacco products. With Ca^2+^ being integral for many physiological responses and processes, it stands to reason that its dysregulation would have negative effects. Rowell and colleagues [62] screened several e-liquid flavors for induced changes in Ca^2+^ homeostasis using human bronchial epithelial cells. Banana pudding flavored e-liquid caused ER Ca^2+^ release and activated SOCE. In particular, long exposures to the e-liquid depleted ER Ca^2+^ stores and inhibited store-operated Ca^2+^ entry, suggesting that e-liquids may alter Ca^2+^ homeostasis through short- and long-term mechanisms. Vaping was also shown to have severe effects on disease and disease severity. For example, our lab [63] utilized a murine model to test the effects of vaping on the severity of the coronavirus-dependent pulmonary disease. By exposing mice to vaped e-liquids followed by infection with Mouse Hepatitis Virus (MHV), a murine tropic virus that produces SARs-like pneumonia in mice, preliminary data indicated an increase in the mortality and pathology of MHV-induced pulmonary infection. This study also suggests a complex role for Ca^2+^ mobilization in inflammation and respiratory disease development in relation to vaping. Additional studies [63] also indicate ALI when mice are exposed to vaped e-liquid. Co-exposure models of vaping and MHV as well as vaping and *Klebsiella pneumonia* also indicate an increase in ALI, i.e., increased lung wet/dry ratio. These increases in wet/dry ratio were significant in co-exposure models. One theory surrounding vaping is that it may trigger M0 macrophage polarization to pro-inflammatory M1 macrophages [64]. These M1 macrophages would then lead to increased lung inflammation as well as play a prominent role in disease progression. Interestingly, this polarization process is also mediated by increases in intracellular Ca^2+^. These data indicate the potential side effects of vaping on health and disease outcomes and underlie the need for awareness of these effects as vaping continues to rise in popularity amongst minors and young adults.

## 6. Pulmonary Disease and Pharmacological Interventions Using Ca^2+^ Signaling Mediators

Recent research suggests Ca^2+^ signaling dysregulation is not only the mechanism of the chronic side effects due to smoking and vaping but also a driving factor in pulmonary disease. These data suggest that targets affected by inhibitors of certain Ca^2+^ channels may serve as novel therapeutic targets for mediating many pulmonary diseases. Ca^2+^ signaling can be inhibited by Ca^2+^ antagonists such as dihydropyridines, phenylalkylamines and benzothiazepines [37]. Pyr6, Pyr10 and 2-APB are other known Ca^2+^ channel antagonists whose effects on Ca^2+^ signaling have been well-studied [65,66]. Pyr6, Pyr10 and 2-APB inhibit store-operated Ca^2+^ entry, non-store operated calcium signaling and IP_3_R, respectively. The antagonist 2-APB has been documented to inhibit IP_3_R-induced Ca^2+^ release [65]. Pyr6 and Pyr10 display marked Ca^2+^ inhibition in TRPC- and ORAI-mediated Ca^2+^ entry, respectively [66]. In the previously mentioned murine model [63], the effects of 2-APB on vaping and dual vaping and MHV exposure were investigated. Mice that received the 2-APB treatment displayed improved respiratory characteristics and function vs the mice only infected with MHV. These Ca^2+^ channel antagonists were also able to significantly decrease inflammatory cytokine production as well as decrease pulmonary inflammation in vape-treated mice receiving either 2-APB or Pyr10. This study suggests that the Ca^2+^ channel antagonists 2-APB, Pyr10 and Pyr6 have potential use as novel therapeutics for pulmonary disease. Table 1 (modified from [67]) lists the various compounds targeting Ca^2+^ signaling components and their mechanism of action. Many other compounds exist but have not yet been tested in relation to pulmonary disease, thus offering a potential future avenue of research focused on investigating these Ca^2+^ signaling mediators.

## 7. Conclusions 

Ca^2+^ dysregulation is involved in the manifestation of several pulmonary diseases [48]. Because of this fact, an understanding of Ca^2+^ signaling mechanisms and the role that Ca^2+^ signaling plays in disease pathology is important. However, relatively little is known about the detailed mechanistic roles of Ca^2+^ in disease. Currently, studies have investigated the potential negative effects of vaping in relation to pulmonary disease. Ca^2+^ signaling pathways have more recently garnered attention as a potential mediator of these negative effects. These studies have demonstrated that vaping results in Ca^2+^ dysregulation, leading to decreases in cell viability and increases in inflammation. However, mechanistic details are lacking and will require further investigation. Finally, while Pyr6, Pyr10 and 2-APB were demonstrated to ameliorate disease prognosis (as described above), the mechanism is not well-understood. These data underly a need for studies to better understand the mechanism of action of vaping upon the lungs and the resultant pathology, which will include deciphering the role that Ca^2+^ dysregulation plays within this disease paradigm.

Future avenues of study will focus on further investigating the mechanistic role of Ca^2+^ signaling dysregulation within the context of diseases that might develop due to either vaping and/or tobacco smoking. Understanding the inflammatory pathways involved via mechanistic dissection with the Ca^2+^ channel antagonists 2-APB, Pyr10 and Pry6 will bolster our appreciation of this field of study and potentially yield novel therapeutic targets for diseases related to Ca^2+^ dysregulation. Furthermore, other Ca^2+^ antagonists exist and must be investigated in relation to pulmonary disease. Another aspect of future studies will focus on investigating the role of M0 macrophage polarization to M1 macrophages as this process was recently theorized to play a role in vaping-dependent inflammatory responses.

## Figures and Tables

**Figure 1 biology-10-01053-f001:**
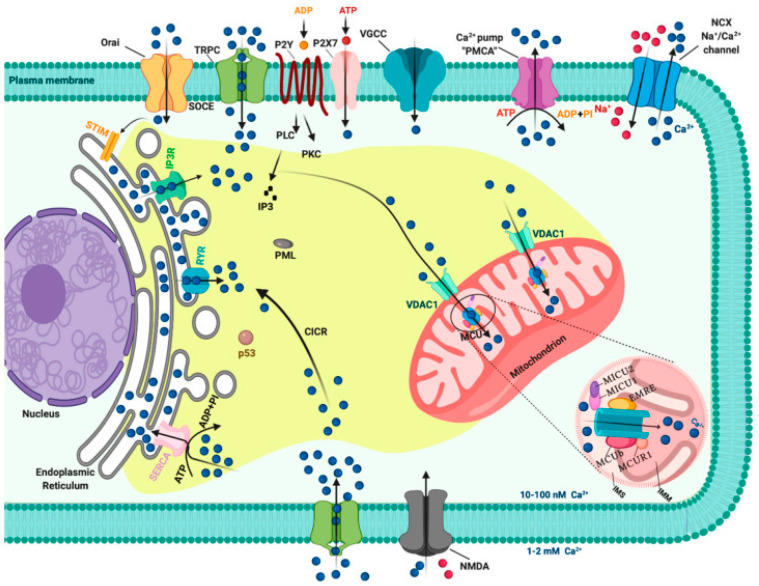
Intracellular calcium (Ca^2+^) signaling. Different Ca^2+^ transporters, channels, exchangers, binding/buffering proteins and pumps mediate the regulation of the cytosolic Ca^2+^ concentration. In the plasma membrane (PM), PM Ca^2+^-ATPases (PMCA) pumps, transient receptor potential channels (TRPC), voltage-gated Ca^2+^ channels (VGCC), Na+/Ca^2+^ exchanger (NCX) and purinergic P2 receptors regulate the transport of Ca^2+^ ions inside and outside cells. Inositol 1,4,5-triphosphate receptors (IP_3_R), ryanodine receptors (RyR) and sarcoendoplasmic reticulum Ca^2+^-ATPase (SERCA) pumps control the storage of Ca^2+^ in the endoplasmic reticulum. Finally, voltage-dependent anion channels (VDAC) and members of the mitochondrial Ca^2+^ uniporter family are critical for controlling the mitochondrial Ca^2+^ uptake. Abbreviations: CICR: Calcium-induced calcium release; PML: Promyelocytic leukemia protein; MICU1: Mitochondrial calcium uniporter component 1; MICU2: Mitochondrial calcium uniporter component 2; EMRE: Essential MCU regulator; MCUb: Calcium uniporter regulatory subunit; IMS: Intermembrane space; IMM: inner mitochondrial membrane. From [20] From Patergnani et al., 2020, with the original work cited as an open access article, thus allowing for use under the terms of the Creative Commons Attribution (CC BY) license (http://creativecommons.org/licenses/by/4.0/ accessed on 26 July 2021).

**Table 1 biology-10-01053-t001:** Compounds targeting Ca^2+^ signaling components.

Ca^2+^ Signaling Components	Pharmacological Compounds	Mechanistic Action
Transient Receptor Potential Channels	Pyr6	Inhibitor
	20-GPPD	Activator
	SKF96365	Blocker
	D-3263	Agonist
	Capsaicin	Agonist
	Cannabidol	Agonist
	SOR-C13 Dexamethasone	Inhibitor
	2-APB	Agonist
Store-operated calcium channels	Pyr10	Inhibitor
	SKF96365	Inhibitor
	DPB-162AE/-163AE ML-9	Inhibitor/Activator
	R02959	Inhibitor
Inositol 1,4,5 triphosphate receptors (IP_3_R)	Heparin, Caffeine	Inhibitor
	Xestospongin B	Inhibitor
	Xestospongin C	Inhibitor
	2-APB	Inhibitor
Voltage-Gated Ca^2+^	Dihydropyridine	Inhibitor

## Data Availability

Not applicable.

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
