# Peer review of "Calcium-Dependent Pulmonary Inflammation and Pharmacological Interventions and Mediators"

_biology, 2021, doi:10.3390/biology10101053_

Round 1
Reviewer 1 Report
The manuscript by Shipman et al. briefly summarizes the field of Ca2+ signaling as it relates to vaping, and inflammatory responses.
Specific comments:
The abstract contains many grammatical errors. Please go through the document and make the necessary changes.
Line 20: change to “leads”.
Line 23: change to “worsening”.
Line 24-25: Line should read, “…actions by which smoking, and vaping may worsen disease”.
Line 28: change to “triggering”.
Line 46: While smoking is certainly the biggest risk factor, there are also environmental and genetic factors that also contribute to the development of pulmonary diseases. The authors should consider rewriting this sentence to reflect this.
Line 90: The “C” in TRPC typically refers to the family group. In this case, the canonical family of TRP channels.
The authors provide a brief introduction into several Ca2+ channels involved in signal transduction. However, the extent to which they describe the channels is probably beyond the scope of their review. The authors should consider trimming it down. For the purpose of this manuscript, a cursory look into the major contributing channels involved is sufficient. The authors need not provide an exhaustive summary of these vascular Ca2+ channels in their work, as there are many comprehensive reviews the authors can refer the readers to for more information (i.e., Brozovich, 2016; Touyz, 2018).
Line 116: Remove “and” from “Ca2+ and release-activated Ca2+”.
Lines 207-208: “Voltage-dependent anion channels”
Author Response
Dear Editor,
I greatly appreciate these inciteful reviews and critiques of our work. We have strived to answer all questions and respond to all comments to our best ability.
Please see responses to comments below:
Reviewer 1:
The manuscript by Shipman et al. briefly summarizes the field of Ca2+ signaling as it relates to vaping, and inflammatory responses.
Specific comments:
The abstract contains many grammatical errors. Please go through the document and make the necessary changes.
Line 20: change to “leads”.
Changed
Line 23: change to “worsening”.
Changed
Line 24-25: Line should read, “…actions by which smoking, and vaping may worsen disease”.
Changed
Line 28: change to “triggering”.
Changed
Line 46: While smoking is certainly the biggest risk factor, there are also environmental and genetic factors that also contribute to the development of pulmonary diseases. The authors should consider rewriting this sentence to reflect this.
Changed to reflect smoking as a major risk factor
Line 90: The “C” in TRPC typically refers to the family group. In this case, the canonical family of TRP channels.
Removed all Cs so that TRP channels are labeled correctly
The authors provide a brief introduction into several Ca2+ channels involved in signal transduction. However, the extent to which they describe the channels is probably beyond the scope of their review. The authors should consider trimming it down. For the purpose of this manuscript, a cursory look into the major contributing channels involved is sufficient. The authors need not provide an exhaustive summary of these vascular Ca2+ channels in their work, as there are many comprehensive reviews the authors can refer the readers to for more information (i.e., Brozovich, 2016; Touyz, 2018). Edited accordingly.
Line 116: Remove “and” from “Ca2+ and release-activated Ca2+”.
Changed
Lines 207-208: “Voltage-dependent anion channels”
Changed

Reviewer 2 Report
The manuscript by Shipman et al. discusses the potential role of intracellular calcium dysregulation and altered calcium signaling in diverse lung diseases. The topic is timely, and it is within the scope of the research community. The cytoplasmic ionic calcium is tightly regulated in eukaryotic cells due to its important role in muscular contraction, neuronal depolarization, and axonal neurotransmitter release and as a secondary messenger in diverse signaling pathways in non-excitable cells. The authors delineate distinct channel, transporters, pumps and exchangers that plays role in calcium movement between intracellular organelles and the cytoplasm or extracellular space and cytoplasm. The manuscript also summarizes the knowledge on smoking and intracellular calcium dysregulation as possible adjuvant pathomechanism behind COPD caused by cigarette smoking and vaping. The authors also speculate on the possible use of the wide variety of agonist and inhibitors that acts on the diverse component of intracellular calcium regulation and signaling.
Main critics:
- Although the authors provide a considerable amount of knowledge on diverse mechanisms that regulate calcium signaling in non-neuronal cells, but mixing in the description of channels that participate in muscular contraction will confuse the reader.
I believe, the manuscript should concentrate either exclusively on epithelial cells or give a wider description of each channel, transporter, etc. for diverse cell types in the lung (epithelial, smooth muscle, granulocytes, lymphocytes, macrophages, etc.).
Choosing either of the options would provide a more cohesive picture.
- The authors tangentially discuss infectious and inflammatory diseases in the lungs but do not specify calcium homeostasis in immune cells( e.g. calcium signaling in lymphocyte activation upon antigen binding). This has to be discussed in more detail.
- Also, asthma as a significant pulmonary disease was not discussed in the manuscript. Calcium-dependent and independent hyperactivation of airway smooth muscle and pharmacological intervention would be of great interest.
- In lines 249-250, the authors state the following: “Studies have demonstrated that smoking induces inflammatory responses in airways and increases the influx of Ca2+ [10, 56].”
Influx is used for calcium uptake from extracellular space into the cytoplasm or it can be used when calcium moves from the cytoplasm to cell organelles (ER, lysosomes, etc.).
Reference 10 states that: “Furthermore, tobacco induced increases in cytoplasmic Ca2+i were not inhibited by (i) the removal of extracellular Ca2+, (ii) endoplasmic reticulum Ca2+ depletion with thapsigargin, or (iii) addition of the mitochondrial uncoupler carbonyl cyanide 3-chlorophenyl hydrazone (CCCP) indicating that Ca2+ influx, endoplasmic reticulum, and mitochondrial Ca2+ stores are not involved in tobacco-induced Ca2+ release. However, this increase in cytosolic Ca2+ was bafilomycin sensitive, suggesting that the Ca2+ came from lysosomes and persisted even with chronic tobacco exposure.”
Also, reference 56 states the following: “ TPM-induced [Ca2+]i increase was not related to extracellular Ca2+ and did not require the activation of the IP3 pathway nor involved the transient receptor potential (TRP) channels. Our findings indicate that, in cells having either intact or depleted endoplasmic reticulum (ER) Ca2+ stores, TPM-mediated [Ca2+]i increase involves cytosolic Ca2+ pools other than thapsigargin-sensitive ER Ca2+ stores.
So, the cytoplasmic calcium level is not the result of calcium influx. Please, correct it.
- Please also, correct the statement in line 252 “However, in contrast, another study [18] demonstrated that Ca2+ influx in the bronchial epithelial cells was decreased in smokers’ lungs, which occurred simultaneously with a decrease in Ca2+ release from the ER as a result of the differential expression of ORAI3.”
Because references 10 and 56 do not state that the calcium influx is increased so the statement in line 252 “in contrast” is not correct.
I believe the review needs to provide more details according to the title and better organization to engage readers.
Author Response
Dear Editor,
I greatly appreciate these inciteful reviews and critiques of our work. We have strived to answer all questions and respond to all comments to our best ability.
Please see responses to comments below:
Reviewer #2:
The manuscript by Shipman et al. discusses the potential role of intracellular calcium dysregulation and altered calcium signaling in diverse lung diseases. The topic is timely, and it is within the scope of the research community. The cytoplasmic ionic calcium is tightly regulated in eukaryotic cells due to its important role in muscular contraction, neuronal depolarization, and axonal neurotransmitter release and as a secondary messenger in diverse signaling pathways in non-excitable cells. The authors delineate distinct channel, transporters, pumps and exchangers that plays role in calcium movement between intracellular organelles and the cytoplasm or extracellular space and cytoplasm. The manuscript also summarizes the knowledge on smoking and intracellular calcium dysregulation as possible adjuvant pathomechanism behind COPD caused by cigarette smoking and vaping. The authors also speculate on the possible use of the wide variety of agonist and inhibitors that acts on the diverse component of intracellular calcium regulation and signaling.
Main critics:
- Although the authors provide a considerable amount of knowledge on diverse mechanisms that regulate calcium signaling in non-neuronal cells, but mixing in the description of channels that participate in muscular contraction will confuse the reader.
I believe, the manuscript should concentrate either exclusively on epithelial cells or give a wider description of each channel, transporter, etc. for diverse cell types in the lung (epithelial, smooth muscle, granulocytes, lymphocytes, macrophages, etc.).
Removed instances of other cell types to focus on the epithelial cell
Choosing either of the options would provide a more cohesive picture.
- The authors tangentially discuss infectious and inflammatory diseases in the lungs but do not specify calcium homeostasis in immune cells( e.g. calcium signaling in lymphocyte activation upon antigen binding). This has to be discussed in more detail.
Added information on calcium homeostasis in Immune cells
- Also, asthma as a significant pulmonary disease was not discussed in the manuscript. Calcium-dependent and independent hyperactivation of airway smooth muscle and pharmacological intervention would be of great interest.
Added description of Asthma when mentioning other pulmonary diseases such as COPD and ARDS
- In lines 249-250, the authors state the following: “Studies have demonstrated that smoking induces inflammatory responses in airways and increases the influx of Ca2+ [10, 56].”
Influx is used for calcium uptake from extracellular space into the cytoplasm or it can be used when calcium moves from the cytoplasm to cell organelles (ER, lysosomes, etc.).
Reference 10 states that: “Furthermore, tobacco induced increases in cytoplasmic Ca2+i were not inhibited by (i) the removal of extracellular Ca2+, (ii) endoplasmic reticulum Ca2+ depletion with thapsigargin, or (iii) addition of the mitochondrial uncoupler carbonyl cyanide 3-chlorophenyl hydrazone (CCCP) indicating that Ca2+ influx, endoplasmic reticulum, and mitochondrial Ca2+ stores are not involved in tobacco-induced Ca2+ release. However, this increase in cytosolic Ca2+ was bafilomycin sensitive, suggesting that the Ca2+ came from lysosomes and persisted even with chronic tobacco exposure.”
Also, reference 56 states the following: “ TPM-induced [Ca2+]i increase was not related to extracellular Ca2+ and did not require the activation of the IP3 pathway nor involved the transient receptor potential (TRP) channels. Our findings indicate that, in cells having either intact or depleted endoplasmic reticulum (ER) Ca2+ stores, TPM-mediated [Ca2+]i increase involves cytosolic Ca2+ pools other than thapsigargin-sensitive ER Ca2+ stores.
So, the cytoplasmic calcium level is not the result of calcium influx. Please, correct it. Corrected.
- Please also, correct the statement in line 252 “However, in contrast, another study [18] demonstrated that Ca2+ influx in the bronchial epithelial cells was decreased in smokers’ lungs, which occurred simultaneously with a decrease in Ca2+ release from the ER as a result of the differential expression of ORAI3.”
Because references 10 and 56 do not state that the calcium influx is increased so the statement in line 252 “in contrast” is not correct.
The section was rewritten to more accurately [present what the references were reflecting.
I believe the review needs to provide more details according to the title and better organization to engage readers.

Reviewer 3 Report
In the present manuscript, the authors aimed at reviewing the role of calcium signaling in the inflammatory pulmonary diseases associated to smoking and vaping, namely ALI, ARDS, COPD and lung cancer. This is an interesting topic and calcium signal is probably strongly involved in such diseases.
The present review is well written but some details should be provided to have a better overview of the involvement of the calcium signaling pathways described at the beginning of the manuscript (sections 2 and 3). Indeed, the authors described TRPC, SOCE, VGCC, PMCA, SERCA and ER and mitochondrial calcium channels but when the authors addressed the smoking and vaping effects they did not mention the role or not of all the calcium channels described earlier. They only indicate an increase and/or a decrease of calcium homeostasis. Moreover, the lung is composed of many cell types therefore the cell type involved in the cited studies should be precised. A diagram or a table describing all the calcium signaling pathways involved according to each lung cell type would improve the manuscript.
On the other hand, the review is related to lung therefore the authors should concentrate on lung and not detail the calcium channels in heart, skeletal muscles, pancreatic beta cells and neurons and calcium dysregulation in neurodegenerative diseases for instance. Similarly, table 1 should only indicate the compounds targeting the calcium channels known to be involved in pulmonary inflammatory diseases associated to smoking or vaping.
The conclusion is usually a summary of the manuscript with the take-home messages and examples should be removed or shifted to other paragraphs (lines 347 to 353).
Abbreviations should be defined when first mentioned (TRPC, SOCE and VGC line 86, VGCC line 152, PM line 123, MHV line 301).
Some abbreviations on the figure 1 are not defined in the legend of the figure (CICR, PML, MICU1, MICU2, EMRE, MCUb, IMS and IMM).
Line 208, change “ion” by “anion” in the definition of VDAC.
Author Response
Dear Editor,
I greatly appreciate these inciteful reviews and critiques of our work. We have strived to answer all questions and respond to all comments to our best ability.
Please see responses to comments below:
Reviewer 3:
In the present manuscript, the authors aimed at reviewing the role of calcium signaling in the inflammatory pulmonary diseases associated to smoking and vaping, namely ALI, ARDS, COPD and lung cancer. This is an interesting topic and calcium signal is probably strongly involved in such diseases.
The present review is well written but some details should be provided to have a better overview of the involvement of the calcium signaling pathways described at the beginning of the manuscript (sections 2 and 3). Indeed, the authors described TRPC, SOCE, VGCC, PMCA, SERCA and ER and mitochondrial calcium channels but when the authors addressed the smoking and vaping effects they did not mention the role or not of all the calcium channels described earlier. They only indicate an increase and/or a decrease of calcium homeostasis. Moreover, the lung is composed of many cell types therefore the cell type involved in the cited studies should be precised. A diagram or a table describing all the calcium signaling pathways involved according to each lung cell type would improve the manuscript
Regarding specified pulmonary cell lines and lung cell: We were unable to create a diagram specifying calcium pathways in relation to lung, as studies did not identify calcium channels.
On the other hand, the review is related to lung therefore the authors should concentrate on lung and not detail the calcium channels in heart, skeletal muscles, pancreatic beta cells and neurons and calcium dysregulation in neurodegenerative diseases for instance. Similarly, table 1 should only indicate the compounds targeting the calcium channels known to be involved in pulmonary inflammatory diseases associated to smoking or vaping.
We removed mention of calcium channels in cell types not related to lung. We removed calcium channels not studied currently in pulmonary diseases in table.
The conclusion is usually a summary of the manuscript with the take-home messages and examples should be removed or shifted to other paragraphs (lines 347 to 353).
Removed and shifted to end of “Emerging tobacco products and calcium derangement and disease” section
Abbreviations should be defined when first mentioned (TRPC, SOCE and VGC line 86, VGCC line 152, PM line 123, MHV line 301).
Added definitions first time abbreviations mentioned
Some abbreviations on the figure 1 are not defined in the legend of the figure (CICR, PML, MICU1, MICU2, EMRE, MCUb, IMS and IMM). (Did acronyms)
Added acronyms to figure legend
Line 208, change “ion” by “anion” in the definition of VDAC.
Changed

Round 2
Reviewer 1 Report
The authors have satisfactorily addressed my concerns. However, there are still some minor issues with grammar and sentence structure remaining. I suggest the authors have the manuscript revised by someone with a solid command of the English language to improve the overall clarity of the manuscript.
Author Response
We appreciated the reviewer’s concerns. We have reviewed paper to improve grammar and sentence structure, and believe that the writing is more clear.
Reviewer 2 Report
The manuscript became more cohesive serving the original goal of authors to depict an overview of intracellular calcium regulation and signaling regarding major pulmonary diseases. The addition of the section that describes the calcium signaling in cellular immune regulation expands our understanding of the role of calcium ions in diverse pulmonary pathologies. The authors corrected the manuscript according to stated in the references as was suggested in the previous review.
Minor critic:
Line 308: Calu3 cells are stated incorrectly as human lung basal epithelial cells. These cells are derived from a patient with non-small-cell lung cancer, so these cells are cancer cells and are our closest model to an airway submucosal glandular epithelial cells. Please correct the statement accordingly.
Author Response
We thank you for your guidance, and the statement has been appropriately corrected.
Reviewer 3 Report
Regarding specified pulmonary cell lines and lung cell: We were unable to create a diagram specifying
calcium pathways in relation to lung, as studies did not identify calcium channels.
response: In that case, why the authors have extensively detailed calcium channels if there is no way to identify them in the studies related to lung ? This part should be reduced and better organised. Authors could only cite the big families of calcium permeable channels (ROC, SOCE, voltage-gated channels which are plasma membrane channels and intracellular calcium channels).
Authors now mention purinergic receptors but there are also other receptor-operated calcium permeable channels (ROC) such as receptors to serotonin and others.
Author Response
We thank the reviewer for their suggestions and have since removed components of first section regarding neuronal and other non-lung Ca2+ signaling, and have reorganized to improve clarity.